# Adapting the Community-based Health Planning and Services (CHPS) to engage poor urban communities in Ghana: protocol for a participatory action research study

Mary Abboah-Offei  ,[1] Akosua Gyasi Darkwa,[2] Andrews Ayim,[2] Adelaide Maria Ansah-Ofei,[3] Delanyo Dovlo,[2] John K Awoonor-Williams,[4] Erasmus Emmanuel Akurugu Agongo,[2] Irene Akua Agyepong,[4] Helen Elsey[1]

¹Department of Health Sciences, University of York, York, UK
²Public Health Faculty, Ghana College of Physicians and Surgeons, Accra, Greater Accra, Ghana
³School of Nursing and Midwifery, University of Ghana, Legon, Greater Accra, Ghana
⁴Health Policy, Ghana Health Service, Accra, Greater Accra, Ghana

**Correspondence to**
Dr Mary Abboah-Offei;
mary.abboah-offei@york.ac.uk

## ABSTRACT

**Introduction** With rapid urbanisation in low-income and middle-income countries, health systems are struggling to meet the needs of their growing populations. Community-based Health Planning and Services (CHPS) in Ghana have been effective in improving maternal and child health in rural areas; however, implementation in urban areas has proven challenging. This study aims to engage key stakeholders in urban communities to understand how the CHPS model can be adapted to reach poor urban communities.

**Methods and analysis** A Participatory Action Research (PAR) will be used to develop an urban CHPS model with stakeholders in three selected CHPS zones: (a) Old Fadama (Yam and Onion Market community), (b) Adedenkpo and (c) Adotrom 2, representing three categories of poor urban neighbourhoods in Accra, Ghana. Two phases will be implemented: phase 1 ('reconnaissance phase) will engage and establish PAR research groups in the selected zones, conduct focus groups and individual interviews with urban residents, households vulnerable to ill-health and CHPS staff and key stakeholders. A desk review of preceding efforts to implement CHPS will be conducted to understand what worked (or not), how and why. Findings from phase 1 will be used to inform and co-create an urban CHPS model in phase 2, where PAR groups will be involved in multiple recurrent stages (cycles) of community-based planning, observation, action and reflection to develop and refine the urban CHPS model. Data will be managed using NVivo software and coded using the domains of community engagement as a framework to understand community assets and potential for engagement.

**Ethics and dissemination** This study has been approved by the University of York's Health Sciences Research Governance Committee and the Ghana Health Service Ethics Review Committee. The results of this study will guide the scale-up of CHPS across urban areas in Ghana, which will be disseminated through journal publications, community and government stakeholder workshops, policy briefs and social media content. This study is also funded by the Medical Research Council, UK.

### Strengths and limitations of this study

► Close engagement with Ghana Health Service and communities throughout the study will enable the development of an urban Community-based Health Planning and Services model that can be delivered sustainably within the current health system.

► Using Participatory Action Research (PAR) will enable increased engagement and the collaboration with research participants and stakeholders.

► The mixed methods used within the PAR approach in three different poor urban neighbourhoods will provide in-depth understanding of the health needs of vulnerable urban residents.

► PAR is time intensive and will require prolonged engagement with the research setting and stakeholders.

► Given the level of engagement involving the study, the COVID-19 pandemic presents a major risk to our ability to implement the study as planned.

## INTRODUCTION

Sub-Saharan Africa is urbanising fast; with over half of the continent's population predicted to live in urban areas by 2035.[1] Governments, at national and city level, face multiple challenges in addressing the growing health needs of their expanding populations.[2] Despite outdated notions of an 'urban advantage', proximity to healthcare does not equate to access to free, high-quality healthcare services or health-promoting interventions for the urban poor.[3] This often results in worse outcomes than their rural counterparts and better-off urban residents.[4] Ghana is one of the most urbanised countries in sub-Saharan Africa with 56.7% of the population estimated to be living in urban areas in 2019.[5] These rapidly expanding cities and towns are

characterised by slum and periurban communities with poor infrastructure, overcrowded and unsanitary conditions, which increase the risk for both communicable and non-communicable diseases, resulting in inequalities, poverty and marginalisation.[2 6] With public services struggling to reach poor urban communities, people turn to a range of predominantly private clinics and pharmacies; however, with limited uptake of health insurance by the poorest,[6] their access to quality and affordable healthcare services and health-promoting activities is severely limited. This has impacted on infant and child mortality which is five times higher in poor urban communities compared with the general urban population.[6] Ensuring appropriate, quality service delivery to households will improve timely, suitable care and health promotion activities and reduce vulnerability to expensive and inappropriate care delivery through a plethora of unregulated providers.

Ghana's three-tier district health system has at its foundation the Community-based Health Planning and Services (CHPS) programme, which has been successfully delivering universal access to health promotion, prevention and basic curative care in rural districts using community-based nurses known as community health officers (CHOs) and volunteers.[7–16] This has led to a reduction in childhood mortality by one-third[17] and decline in total fertility by one birth.[18] However, despite government policy to scale up CHPS nationally, these benefits do not currently extend to the urban population.[19 20] CHPS implementation in urban areas of Ghana has been limited to a few pilot districts[7] and there are calls for more research to inform an urban CHPS model.[8] Evidence from CHPS piloted in some urban areas has revealed a need for greater range of services including improved approaches to the delivery of the Integrated Management of Childhood Illnesses; sustained and expanded engagement of communities and volunteering programmes; improved motivation and skills training for staff and opportunities for career progression.[20] Limited registration with Ghana's National Health Insurance Scheme (NHIS) further undermines access to health services in urban areas.[21] NHIS was created in 2003 with the ideal of being 'pro-poor', however, recent studies have shown only 17% of the poor are insured compared with 44% among rich households.[22] Although those classified as extremely poor automatically qualify for a free NHIS card,[23] the extent to which this is taken up in poor urban areas is unknown. Developing appropriate and sustainable system-wide solutions in the urban context is vital if urban CHPS is to move beyond a few pilot areas.

Community engagement is a central pillar of CHPS with six milestones of implementation[8]; however, in many poor urban areas, social structures and cohesion look very different from those found in rural areas. Poor urban neighbourhoods are frequently characterised by transient migrant populations, with both men and women working long hours with reduced support from extended family and multiple stresses of urban living.[24]

Studies have identified particular challenges in engaging urban communities in health programmes and activities.[25–27] There are many examples of community engagement approaches, particularly in rural areas, such as the social accountability approach which has been described by the WHO as a method of community engagement that brings relevant perspectives together in conversations where everyone is an equal partner.[28] Other key examples of community engagements that have led to successful roll out of community clinics targeting poor urban populations are those of the Mohalla and Basthi Dawakhana clinics in India.[29 30] Understanding how best to engage poor urban households, particularly the most vulnerable women and children, is vital for effective adaptation and scale up of CHPS within urban areas. Methods for community engagement in decisions that affect their lives have been criticised for the lack of participatory approaches and inability to move beyond tokenistic mode of participation.[31–33]

One approach to addressing these criticisms is Participatory Action Research (PAR),[34 35] which will help to identify and specify system adaptations required to effectively implement CHPS sustainably to reach the urban poor. The PAR approach will enable meaningful engagement with stakeholders, communities and vulnerable households in identifying social structures, health needs and other areas of concern through collaboration and capacity building to address these issues holistically.[36–38] In order to enhance a better understanding of engagement within health systems, this study has adopted the conceptual framework derived from a systematic review of public health interventions (see figure 1).[39] This framework provides clarity on aspects of engagement from a community and a health system perspective, and has the advantage of being an empirically driven model using findings from both qualitative and quantitative studies.[39] Although studies included in this review were predominantly from high-income countries, the insights show similarities to reviews of engagement in low/middle-income countries (LMICs).[25–27]

Therefore, this study aims to engage CHPS team and managers, volunteers, community members and key stakeholders to understand their social structures, health needs, health seeking and health insurance behaviour, and to identify and specify system adaptations required to effectively implement CHPS sustainably and at scale to reach the urban poor in Ghana.

### Study objectives
1. To explore social structures of poor urban communities, and vulnerable individuals and households, including the uninsured, and identify their current health-seeking behaviour, using this information to adapt the CHPS model.
2. To review and critically analyse preceding efforts to implement rural and urban CHPS to understand what worked (or not), how and why.

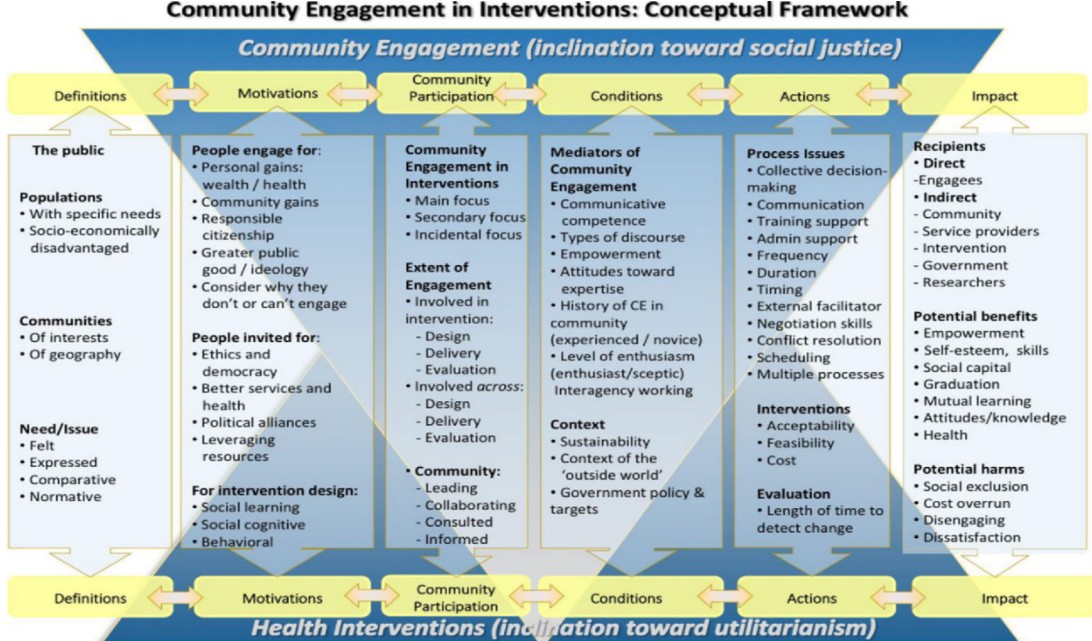

**Figure 1** A conceptual framework for community engagement in interventions, Brunton *et al.*[39]

3. To design with urban communities, CHPS community teams and health system managers an adaptation of the current CHPS programme and system to ensure urban relevance.
4. To evaluate early implementation processes, costs, and acceptability from the perspective of urban populations who are marginalised and vulnerable such as women and children and frontline health workers, volunteers and health systems managers, and identify the pathway for scale up.
5. To make recommendations on contextually appropriate modifications to the CHPS and NHIS policy, programme and implementation arrangements for urban localities as well as approaches to scale up.

## METHODS
### Study design
The study uses both quantitative and qualitative methods within the overall approach of PAR. PAR is a rigorous and systematic approach to enquiry, which allows researchers and stakeholders to explore and discover effective solutions to life problems.[40] PAR gives stakeholders the opportunity to be involved in multiple recurrent stages (cycles) of community-based planning, action, observation and reflection[41] with each cycle following on from and influencing subsequent cycles.[40 42] The use of PAR in the community is beneficial in increasing engagement and the collaborative nature of the study.[43 44] We chose PAR as an appropriate methodology for this study as it will enable us to try out different approaches to engagement in a range of poor urban settings and reflect on the experience in collaboration with CHOs, volunteers and community members.

In addition, given that community engagement is a key component of CHPS, the alignment of PAR to community engagement will help to strengthen CHPS staff skills in participation approaches, while simultaneously strengthening engagement bonds with poor urban communities. This level of engagement we hope will increase community ownership of the urban CHPS model, supporting scale-up and sustainability. For example, we will clarify, from the perspective of communities and CHPS, which population groups and communities should be targeted and understand the motivations, both from communities and CHPS, for their inclusion. This will include understanding the motivations for registering (or not) for NHIS and facilitating those excluded from NHIS to identify ways to encourage registration among these groups. We will also explore the extent of participation, the mediators and context for engagement. Our PAR cycles will allow us to consciously address process issues and document the successes for different approaches to process. We will explore the impact of community engagement on participants, the wider community, within the CHPS teams and the health system (both potential benefits and harms). This study will be implemented in two phases: phase 1 will address study objectives 1–2, which will constitute 'reconnaissance phase' in PAR (see figure 2); and phase 2 will address study objectives 3–5 and will comprise a number of PAR cycles (see figure 3).

### Setting
We purposively selected three poor urban CHPS zones: (a) Old Fadama (Yam and Onion Market community), (b) Adedenkpo and (c) Adotrom 2 that do not currently have functional CHPS programme (ie, a designated CHPS zone that is lacking in facilities, staff or targeted outreach

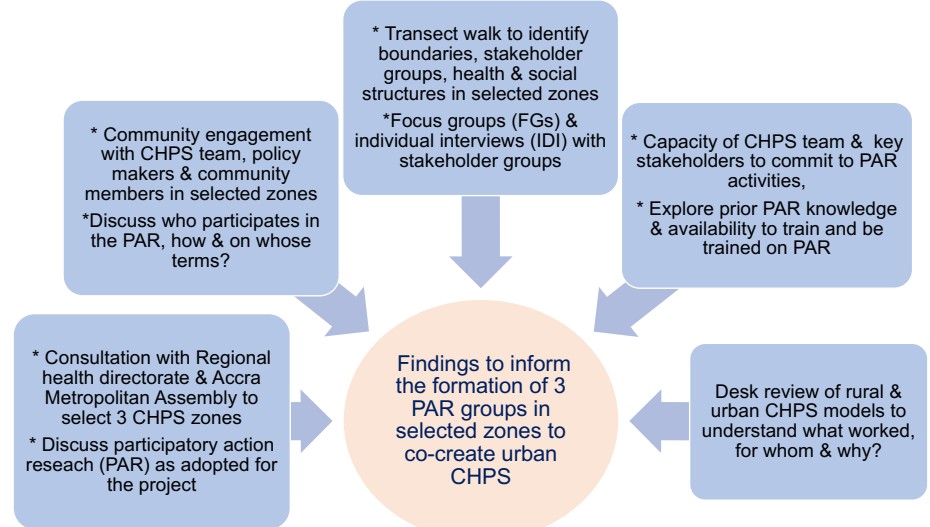

**Figure 2** Participatory Action Research (PAR) activities in reconnaissance phase. CHPS, Community-based Health Planning and Services.

services), and with differing characteristics, to allow for transferability across urban areas of differing social characteristics. A CHPS zone includes about 5000–10 000 population assigned to a Community Health Team led by a CHO. CHPS zones are linked to primary health clinics/ maternity homes as the next level of care, and then to

District/Municipal Hospital and Health Management Team. This three-tier system allows for referral as well as supervision and monitoring from the community level. The three selected zones will include: (a) an informal settlement of predominantly first-generation migrants; (b) a mixed poor/better-off neighbourhood and (c) a

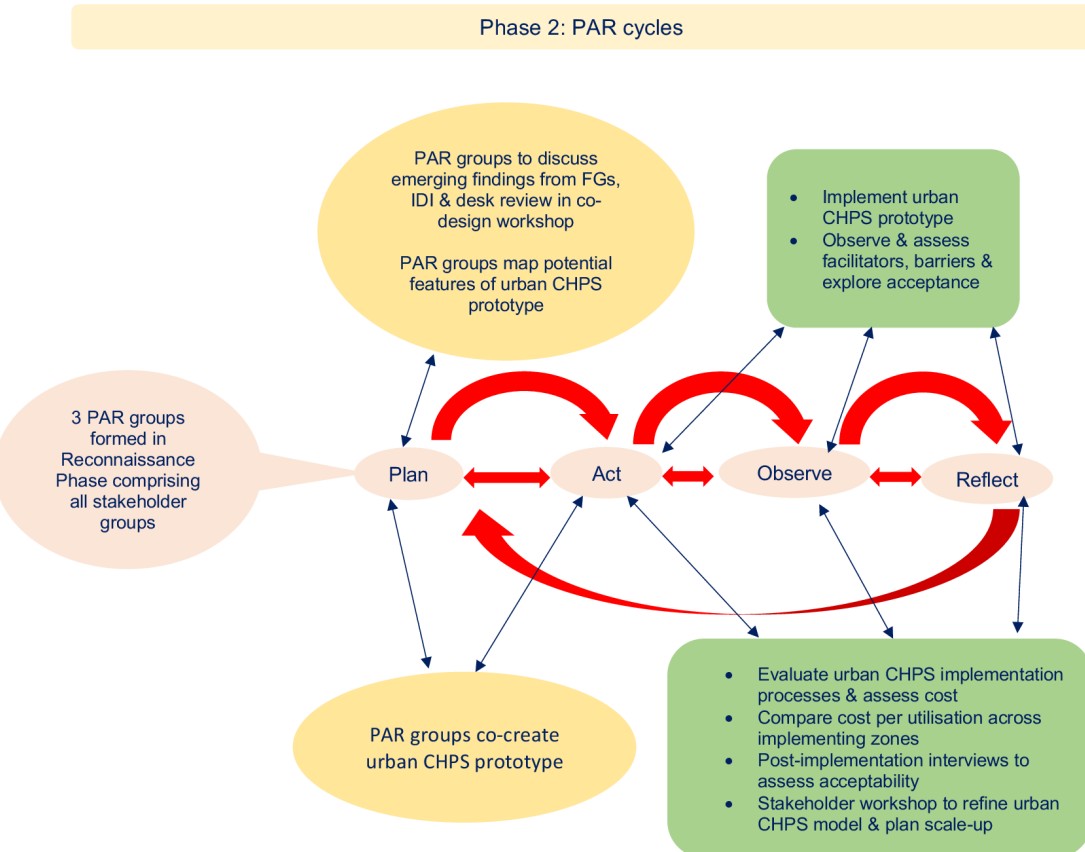

**Figure 3** Proposed PAR cycles and activities. CHPS, Community-based Health Planning and Services; FGs, focus groups; IDI, individual interview; PAR, Participatory Action Research.

long-established neighbourhood of several generations. We will also select different levels of CHPS functionality (eg, with facilities and staff but no volunteers or outreach, or with no facilities or staff in the demarcated zone but with support from neighbouring zones).

## Study phase 1

### Reconnaissance phase

The 'reconnaissance phase'[45] is described in the action research literature as an observational phase to gain insight into the problem and develop a theorised account to inform action.[45] The information generated in the reconnaissance phase will inform and enable the establishment of groups of co-researchers in each of the three CHPS zones. The findings of the reconnaissance phase will also inform the CHPS models to be tried in the first PAR cycle in each CHPS zone. PAR groups will be made up of key stakeholders, or co-researchers, affected by the health and access problems experienced in their neighbourhoods. The co-research groups will include community members, CHPS staff, volunteers and managers, policymakers and representatives from the regional health directorate. The reconnaissance phase enables theory building within action research,[45] emerging theories will be structured according to the theoretical framework in figure 1.[39] We will also be mindful of emerging aspects not reflected in the framework; this is particularly important given the limited LMIC-based studies that have informed this framework.

This phase will focus on consultations with Greater Accra Regional Health Directorate and Regional/District CHPS coordinators and the Accra Metropolitan Assembly to select three CHPS zones for the study; engagement and transect walks with the local team (CHPS team and managers, community members and other key stakeholders) in each zone; checking and building capacity of community members, key stakeholders and CHPS staff to commit to the study in terms of knowledge on PAR; availability to attend training on PAR or train others to become PAR co-researchers. This phase will address research objectives 1 and 2, enabling us to find a shared concern among key stakeholders and community members in their respective CHPS zones to begin to identify solutions from the perspective of those most affected,[42] particularly community members vulnerable to ill-health and have poor access to health services and CHPS staff and volunteers (see figure 2 for activities in the reconnaissance phase).

For objective 2, we will conduct a mixed-methods review using a results-based convergent design,[46] where the quantitative and qualitative findings are synthesised separately and then brought together in a final synthesis. This will allow us to synthesise quantitative results from included studies on the outcomes of CHPS and findings from qualitative, mixed-methods or quantitative studies on the mechanisms (eg, health system, participant or contextual factors) that may influence effectiveness. We will include any study evaluating the CHPS programme in urban or rural Ghana and will particularly look to identify facilitators and barriers to success in urban contexts. We will also establish the groups of co-researchers in each CHPS zone, share the findings from the reconnaissance work and facilitate discussions on adaptations of the CHPS model to respond to community needs.

### Recruitment and data collection in phase 1

A purposive sample of key health providers and community stakeholders will be recruited from the three selected zones for an initial focus group (FG) discussion. Due to power play and political factions that exist in selected communities, stakeholder groups (policymakers, regional health directorate, CHPS staff and community members) will be engaged separately in FGs and individual interviews prior to engaging all stakeholder groups together in the PAR groups. This will allow for the exploration of perspectives and issues within respective communities, where community members can talk freely without fear of intimidation from other stakeholders. Then during the PAR group meeting, issues raised in the FGs and interviews will be brought to the table where everyone will discuss in an environment of respect and equity.[28] Two FGs each will be conducted in each zone using techniques such as 'social mapping' to facilitate discussion on social structures and health-seeking behaviour, and 'chapati diagrams'[47] to identify health providers and their relative importance (each FG will comprise of 8–10 participants, and a total of n=6 FGs will be held). Local gatekeepers from the FGs will help researchers to identify particularly vulnerable household (eg, with under-5 children, female headed, elderly or chronically ill or disabled) for individual interviews to explore the challenges they face in registering for insurance, keeping healthy and seeking care. Eight individual interviews will be conducted in each zone making a total of n=24 interviews informing 'definitions' and 'motivations' within the framework.[39] The sequential follow-up of individual interviews is aimed at gaining a deeper understanding of the internal and external issues and concerns surrounding the implementation and use of CHPS services.

Data will be collected on baseline costs and utilisation, disaggregated by gender and age, of any elements of from existing urban CHPS pilot. The findings of the mixed-methods systematic review of the CHPS programme in rural and urban areas will be used to inform key informant individual interviews with CHPS staff and volunteers (n=10) and community members (n=10) in the three selected zones. These interviews will explore all six aspects of the framework[39] in figure 1.

### Data analysis for phase 1

All interviews and FGs will be audio recorded and transcribed as soon as data collection is over. Transcripts will be translated into English. Data collected will be reviewed and coded using the Brunton et al[39] domains of community engagement as a framework to understand community assets and potential for engagement.

The data collected in phase 1 will also enable identification of the needs, gaps, weaknesses and opportunities within the three selected zones relating to CHPS, staff and volunteers, community members and other relevant stakeholders. We will follow the seven stages within framework approach as described by Gale *et al*.[48] Data will be managed using NVivo software. This analysis will inform the initial CHPS models considered by the co-researchers in each zone.

Throughout the study, the research team will keep a comprehensive reflective research journal, which will be used to catalogue the progress, obstacles and successes of the PAR process. This journal will be kept to acknowledge the research team's experience of PAR in the urban context, analysis and interpretation.[49 50] The journal will also act as a component of the audit trail for the study.[51] Reflective journals also increase external validity by making subjective processes transparent.[50] During this phase, the CHPS staff, volunteers and researchers will receive training and support on participatory methods and the principles underpinning PAR participation: 'Action research is only possible with, for and by persons and communities, ideally involving all stakeholders both in the questioning and sense making that informs the research, and in the action which is its focus'[52]; and produce practical knowledge, and to do that draws on representational, relational and reflective knowledge and is for a worthwhile purpose.[52]

### Study phase 2: PAR cycles

During PAR cycles in study phase 2, the three groups of co-researchers will work through the cycles of 'plan, act, observe, reflect'[34] in their respective CHPS zones, developing the model. The PAR co-researchers (PAR groups) will work through PAR cycles to co-create the urban CHPS model with stakeholders, implement and evaluate the processes, cost and acceptability of this model from the perspectives of the poor urban population. To initiate this process, stakeholder meetings will be held with the PAR groups in each of the three zones to co-create an urban CHPS model drawing on the findings from phase 1. The emerging findings from phase 1 in relation to the conceptual framework[39] adopted (see figure 1) will be discussed with the PAR groups to trigger and inform the design of the prototype urban model. Different design options will be costed to enable consideration of sustainability and scale-up at this early stage. We will also work closely with Ghana Health Service (GHS) and the CHPS programme, taking an embedded research approach[53] to ensure that issues of sustainability and scale-up are central to the design.

Following from the workshop with the PAR groups from the three zones, the co-researchers in each zone will begin the PAR cycles. The PAR groups will be facilitated by public health professionals in training (residents) from the Ghana College of Physicians and Surgeons (GCPS), who will also document the decisions taken by the PAR groups and support them to identify the most effective

methods for assessing whether the adaptations to the CHPS model work in practice in increasing access, feasibility and appropriateness of CHPS in their community. This observation stage will include use of routine health information data to understand patterns of utilisation of CHPS services disaggregated by gender, age and diagnosis (variables routinely available in the District Health Information Management Systems (DHIMS)). The PAR groups will monitor utilisation data on an ongoing basis identifying those excluded and responding by reshaping aspects of the model in subsequent PAR cycles (see figure 3 for PAR cycles and activities in phase 2).

### Data collection in phase 2

The co-researchers and public health residents will use qualitative methods including observation, interviews and FGs as appropriate to understand issues of acceptability, feasibility and access in more depth and from the perspective of those most likely to struggle in accessing healthcare and improving health. Actual costs of delivering the model from the health service perspective in the different urban zones will be collected. This will include costs of staff and volunteer training, time and salary or expenses, transportation, materials and equipment, and supervision. The utilisation data will be used to estimate the cost per patient in each of the three zones. Emerging findings from the co-researchers in the three zones will be discussed with GHS and CHPS programme managers over the 12-month period of implementation of the PAR cycles. At the end of the 12 months, a workshop will be held with all stakeholder groups (CHPS managers and staff, provincial and national health system stakeholders, policymakers and community members) to assess the findings from the selected zones in order to plan the next steps in implementation and scale-up, if appropriate. We will develop guidelines, training and standard operating procedures, recording and reporting formats for the urban CHPS model based on study findings (see figure 4 for all study activities in phases 1 and 2).

### Patient and public involvement

Community members from the selected CHPS zones will be involved throughout the study. The community leaders and the CHPS managers in these selected communities will assist in negotiating access to the community. This will include local recruitment of PAR co-researchers and hosting of series of community engagement meetings with PAR groups where the prototype urban CHPS model design will be discussed before the results are disseminated.

### Rigour

Appropriate measures will be implemented to increase the rigour of this study. Data will be collected and coded by the research team and discussions held regularly with stakeholders to reduce bias.[54] Sources of potential bias that could influence the processes of data collection and analysis due to existing networks and connections will

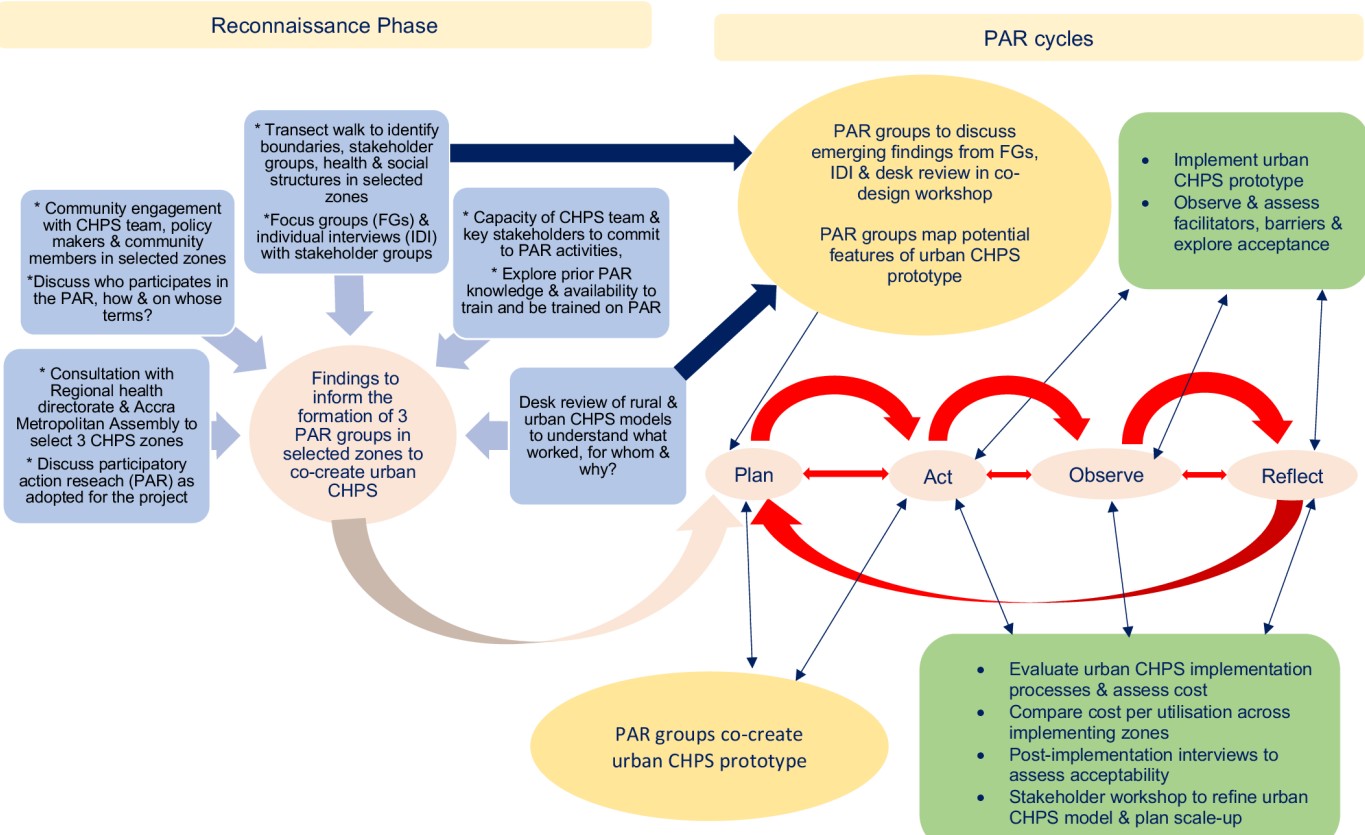

**Figure 4** Proposed Participatory Action Research (PAR) activities and cycles for the whole study. CHPS, Community-based Health Planning and Services.

be acknowledged and recorded. This level of documentation will increase confirmability by providing an audit trail, which will allow observers to confirm the veracity of the study.[40 55] Prolonged engagement with the community and stakeholders will increase credibility and regular member checking of raw data, analyses and reports.[40] Transferability will be provided through detailed descriptions of the contextual data and activities of the study; through immersion, reflective journaling and detailed documentation, this will allow other researchers to analyse the situation and study outcomes based on context.[55 56]

The coding and themes will be analysed by at least two to three members of the research team to enhance credibility.[57] This will involve a reflective practice whereby the team leader will first code the data, and then these codes will be discussed by the research team and further refined to ensure the codes fit with the framework and any emerging themes reflect the dataset. This process will enhance dependability and intercoder reliability.[57] The research team will be involved in the development of all interview guides and further refinement of the guide will occur as a team.

## ETHICS APPROVAL
This study has been approved by the University of York's Health Sciences Research Governance Committee (HSRGC/2020/409/E) and the GHS Ethics Review Committee (GHS-ERC 003/10/20). Given the community participatory nature of this study, there are ethical considerations in relation to protecting the anonymity of participants and confidentiality of data, particularly regarding interviews and FGs. The connected nature of community members will be acknowledged in consent forms, ground rules of confidentiality will be agreed in all group discussions, care will be taken in analysis and presentation of data in ensuring that participant confidentiality is protected. Data that may overtly identify participants will be excluded.[58] Consent will be required from all participants prior to their participation in the study.

## Capacity building and dissemination
This study aims to develop a centre of excellence for training and build capacity for urban CHPS, which will develop and deliver training for CHPS urban teams on clinical and participatory techniques and processes. Under guidance of the local research team in Ghana, public health registrars training under the GCPS will be seconded to the centre evaluating, developing and supporting CHPS throughout scale-up. The PAR process will build capacity of CHPS workers in participatory approaches and use of DHIMS data to develop strategies to reach those who may struggle to access CHPS services.

The results of this study will guide the scale-up of CHPS across urban areas in Ghana providing detailed

information on all components, costs and impact on utilisation, with outputs such as training materials, operational guidelines and policy revisions. This will lay the foundation for a community-driven model that fits sustainably within the GHS. The centre of excellence will facilitate further evaluation and sharing of good practice. Beyond Ghana, this study, and the planned follow-on evaluation of scale-up, will provide much needed evidence and insight into how to engage communities in urban areas so their needs are addressed appropriately by the health system. Policymakers, practitioners and researchers involved in urban health across LMICs are grappling with these issues and searching for solutions to improve the health of the poorest urban residents.

**Acknowledgements** We acknowledge all authors for their contribution to this protocol. We also acknowledge the University of York, Ghana College of Physicians and Surgeons and the Ghana Health Service for supporting this study. Finally, we acknowledge the Medical Research Council for funding this study.

**Contributors** EEAA, HE, IAA and JKA-W developed the concept for the study and successfully sought funds from Medical Research Council to conduct the study. MA-O developed and drafted the initial manuscript. AGD, AA, AMA-O and DD reviewed the initial draft of the manuscript and added to the methodology, plans for analysis and dissemination. All authors reviewed the final draft of the manuscript and approved the final version.

**Funding** The Medical Research Council (grant number MR/T022787/1) supports this work.

**Competing interests** None declared.

**Patient and public involvement** Patients and/or the public were involved in the design, or conduct, or reporting, or dissemination plans of this research. Refer to the Methods section for further details.

**Patient consent for publication** Not required.

**Provenance and peer review** Not commissioned; externally peer reviewed.

**ORCID iD**
Mary Abboah-Offei http://orcid.org/0000-0002-9738-878X

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
