## [Reviewer comments · BMJ Open]

ARTICLE DETAILS

TITLE (PROVISIONAL)	Adapting the community-based health planning and services (CHPS) to engage urban poor communities in Ghana: protocol for a participatory action research study.
AUTHORS	Abboah-Offei, Mary; Gyasi, Akosua; Ayim, Andrews; Ansah-Ofei, Adelaide; Dovlo, Delanyo; Awoonor-Williams, John; Agongo, Erasmus; Agyepong, Irene; Elsey, Helen

VERSION 1 – REVIEW

REVIEWER	Snadden, David The University of British Columbia Faculty of Medicine, Family Practice
REVIEW RETURNED	19-Mar-2021

GENERAL COMMENTS	Thank you for the opportunity to review this protocol. This is a highly ambitious and complex proposal and is very clearly laid out and described. Essentially this project will use Participatory Action Research methods to bring learnings from successful community health planning efforts in rural Ghana to poor urban areas. The background, introduction and conceptual framework are well described and clearly written. The PAR methods are described and the processes for stakeholder engagement and data collection, both quantitative and qualitative, are described. The quantitative data will be drawn from existing data held within the health system. The methods of data analysis in respect of the PAR are laid out and as PAR is an action research process one can expect in the study some adaptation of design and data collection and analysis as the project develops and community voices are heard. All the stages of the project are well thought through and methodologically sound from my perspective. While I am impressed at the complexity and thought behind this project and would have no concerns about it proceeding as described I do have some thoughts which you might want to consider in terms of the process of engagement. This project does fit well under a social accountability umbrella in terms of trying to find ways to improve health in disadvantaged populations. From that perspective the World Health Organisation in its publication "Towards Unity for Health" (World Health Organization. Towards unity for health: challenges and opportunities for partnership in health development: a working paper / Charles Boelen. Geneva, Switzerland: World Health Organization 2000) describes a method of community engagement that brings relevant perspectives together in conversations where everyone is an equal partner. This five health partner approach includes: policymakers, health system managers, healthcare providers, academics, and community members. The main process being that these groups meet
---

	together so everyone can bring perspectives to the table and everyone can hear each other in an environment of respect and equity. This model has been adapted with success in rural British Columbia to add in relevant business and linked sectors (eg non-profits) while recognizing that the appropriate groups differ in every community - the WHO model is a framework for engaging communities equitably through a social accountability lens. A practical example of this approach being used can be found at: https://rccbc.ca/wp-content/uploads/2020/08/BC-Rural-and-First-Nations-Health-and-Wellness-Summit-Summary-Report.pdf in that example a major workshop was held to identify rural community priorities in terms of health care improvement. two further references of interest may be: Woollard RF. Caring for a common future: medical schools' social accountability. Medical Education 2006;40(4):301-13. doi: 10.1111/j.1365-2929.2006.02416.x Agarwal, S., R. Heltberg, and M. Diachok. Scaling-up Social Accountability in World Bank Operations. Washington, DC: World Bank 2009 I offer these reflections as much community engagement has been somewhat hierarchical so that communities don't always feel empowered, and PAR can sometimes turn out that way if one is not careful, therefore I am wondering if there is an opportunity at the beginning stages of your project to engage the community perspectives together in an equitable and socially accountable manner. While you do describe stakeholder focus groups what I am wondering is whether you could bring all the perspectives together in the groups, or use a community workshop approach to get the project started. As you know the context of your research and I don't then I would leave it to you to use an approach that would likely work in your setting.
--	--

REVIEWER	Lahariya, Chandrakant G. R. Medical College, Community Medicine
REVIEW RETURNED	02-Apr-2021

GENERAL COMMENTS	It is a very relevant and useful manuscript, which will help countries to understand how to tackle urban healthcare needs in time ahead. A few parts of this manuscript can be edited and revised significantly. Specially, the sections which describe the study phases, instead of text as paragraphs, the flow charts and simplified figures would add value. Figure 2 is too text heavy. It need to be edited and simplified. Lately, there has been some useful work from countries such as India, which have rolled out community clinics (Mohalla Clinics and Basthi Dawakhana) and those works could be useful reference points for authors.
--

VERSION 1 – AUTHOR RESPONSE

Reviewer: 1 comments to the Author:

Thank you for the opportunity to review this protocol. This is a highly ambitious and complex proposal and is very clearly laid out and described. Essentially this project will use Participatory Action Research methods to bring learnings from successful community health planning efforts in rural Ghana to poor urban areas. The background, introduction and conceptual framework are well described and clearly written. The PAR methods are described and the processes for stakeholder engagement and data collection, both quantitative and qualitative, are described. The quantitative data will be drawn from existing data held within the health system. The methods of data analysis in respect of the PAR are laid out and as PAR is an action research process one can expect in the study some adaptation of design and data collection and analysis as the project develops and community voices are heard. All the stages of the project are well thought through and methodologically sound from my perspective. While I am impressed at the complexity and thought behind this project and would have no concerns about it proceeding as described I do have some thoughts which you might want to consider in terms of the process of engagement. This project does fit well under a social accountability umbrella in terms of trying to find ways to improve health in disadvantaged populations. From that perspective the World Health Organisation in its publication "Towards Unity for Health" (World Health Organization. Towards unity for health: challenges and opportunities for partnership in health development: a working paper / Charles Boelen. Geneva, Switzerland: World Health Organization 2000) describes a method of community engagement that brings relevant perspectives together in conversations where everyone is an equal partner. This five health partner approach includes: policymakers, health system managers, healthcare providers, academics, and community members. The main process being that these groups meet together so everyone can bring perspectives to the table and everyone can hear each other in an environment of respect and equity. This model has been adapted with success in rural British Columbia to add in relevant business and linked sectors (eg non-profits) while recognizing that the appropriate groups differ in every community - the WHO model is a framework for engaging communities equitably through a social accountability lens. A practical example of this approach being used can be found at: <https://rccbc.ca/wp-content/uploads/2020/08/BC-Rural-and-First-Nations-Health-and-Wellness-Summit-Summary-Report.pdf> in that example a major workshop was held to identify rural community priorities in terms of health care improvement.

Two further references of interest may be:

Woollard RF. Caring for a common future: medical schools' social accountability. *Medical Education* 2006;40(4):301-13. doi: 10.1111/j.1365-2929.2006.02416.x

Agarwal, S., R. Heltberg, and M. Diachok. *Scaling-up Social Accountability in World Bank Operations*. Washington, DC: World Bank 2009

I offer these reflections as much community engagement has been somewhat hierarchical so that communities don't always feel empowered, and PAR can sometimes turn out that way if one is not careful, therefore I am wondering if there is an opportunity at the beginning stages of your project to engage the community perspectives together in an equitable and socially accountable manner. While you do describe stakeholder focus groups what I am wondering is whether you could bring all the perspectives together in the groups, or use a community workshop approach to get the project started. As you know the context of your research and I don't then I would leave it to you to use an approach that would likely work in your setting.

Author response:

Thanks for your supportive comments and the additional references suggested. We found the references of relevance to our work and have included the key reference in the introduction. See paragraph 3 on page 3 of the main manuscript.

Thanks the reflections shared on the approaches to community engagement. The manuscript have been revised and updated as ‘Due to power play and political factions that exist in selected communities, stakeholder groups (policy makers, regional health directorate, CHPS staff and community members) will be engaged separately in focus groups and individual interviews prior to engaging all stakeholder groups together in the PAR groups. This will allow for the exploration of perspectives and issues within respective communities, where community members can talk freely without fear of intimidation from other stakeholders. Then during the PAR groups meeting, issues raised in the focus groups and interviews will be brought to the table where everyone will discuss in an environment of respect and equity ²⁸’, see paragraph 2 on page 6 of the main manuscript.

Reviewer: 2 Comments to the Author

It is a very relevant and useful manuscript, which will help countries to understand how to tackle urban healthcare needs in time ahead. A few parts of this manuscript can be edited and revised significantly. Specially, the sections which describe the study phases, instead of text as paragraphs, the flow charts and simplified figures would add value. Figure 2 is too text heavy. It need to be edited and simplified.

Author response:

Thanks for your supportive comments. The study phases have been revised and figures 2 and 3 have been added to the main manuscript. The original Figure has now been revised and updated as figure 4 (see Figures 2, 3 and 4 below).

Figure 2: PAR activities in Reconnaissance Phase

Phase 2: PAR cycles

Figure 3: Proposed PAR cycles and activities

1

Figure 4: Proposed PAR activities and cycles for study Phases 1 & 2

1

Reviewer: 2 Comments to the Author

Lately, there has been some useful work from countries such as India, which have rolled out community clinics (Mohalla Clinics and Basthi Dawakhana) and those works could be useful reference points for authors.

Author response:

Thanks for your comments and the additional key references suggested. The introduction section of the manuscript have been revised and updated as 'Other key examples of community engagements that have led to successful roll out of community clinics targeting urban poor populations are those of the Mohalla and Basthi Dawakhana clinics in India ^{32,33}', see paragraph 3 on page 3 of the main manuscript.

Thank you for the opportunity to revise this manuscript, and please do not hesitate to contact me if you require any further information.